# The Leading Role of the Immune Microenvironment in Multiple Myeloma: A New Target with a Great Prognostic and Clinical Value

**DOI:** 10.3390/jcm11092513

**Published:** 2022-04-29

**Authors:** Vanessa Desantis, Francesco Domenico Savino, Antonietta Scaringella, Maria Assunta Potenza, Carmela Nacci, Maria Antonia Frassanito, Angelo Vacca, Monica Montagnani

**Affiliations:** 1Department of Biomedical Sciences and Human Oncology, Pharmacology Section, University of Bari Aldo Moro Medical School, 70124 Bari, Italy; francescodsavino@gmail.com (F.D.S.); antoscari13@gmail.com (A.S.); mariaassunta.potenza@uniba.it (M.A.P.); carmela.nacci@uniba.it (C.N.); monica.montagnani@uniba.it (M.M.); 2Unit of General Pathology, Department of Biomedical Sciences and Human Oncology, University of Bari Aldo Moro Medical School, 70124 Bari, Italy; antofrassanito@gmail.com; 3Unit of Internal Medicine and Clinical Oncology, Department of Biomedical Sciences and Human Oncology, University of Bari Aldo Moro Medical School, 70124 Bari, Italy; angelo.vacca@uniba.it

**Keywords:** multiple myeloma, bone marrow niche, immune escape, immune exhaustion, immune checkpoint inhibitors, immune microenvironment, immunotherapy

## Abstract

Multiple myeloma (MM) is a plasma cell (PC) malignancy whose development flourishes in the bone marrow microenvironment (BMME). The BMME components’ immunoediting may foster MM progression by favoring initial immunotolerance and subsequent tumor cell escape from immune surveillance. In this dynamic process, immune effector cells are silenced and become progressively anergic, thus contributing to explaining the mechanisms of drug resistance in unresponsive and relapsed MM patients. Besides traditional treatments, several new strategies seek to re-establish the immunological balance in the BMME, especially in already-treated MM patients, by targeting key components of the immunoediting process. Immune checkpoints, such as CXCR4, T cell immunoreceptor with immunoglobulin and ITIM domains (TIGIT), PD-1, and CTLA-4, have been identified as common immunotolerance steps for immunotherapy. B-cell maturation antigen (BCMA), expressed on MMPCs, is a target for CAR-T cell therapy, antibody-(Ab) drug conjugates (ADCs), and bispecific mAbs. Approved anti-CD38 (daratumumab, isatuximab), anti-VLA4 (natalizumab), and anti-SLAMF7 (elotuzumab) mAbs interfere with immunoediting pathways. New experimental drugs currently being evaluated (CD137 blockers, MSC-derived microvesicle blockers, CSF-1/CSF-1R system blockers, and Th17/IL-17/IL-17R blockers) or already approved (denosumab and bisphosphonates) may help slow down immune escape and disease progression. Thus, the identification of deregulated mechanisms may identify novel immunotherapeutic approaches to improve MM patients’ outcomes.

## 1. Introduction

Multiple myeloma (MM) is a neoplastic plasma cell (PC) disorder characterized by clonal proliferation of malignant PCs in the bone marrow microenvironment (BMME). The abnormal and uncontrolled proliferation of PCs translates into the accumulation of monoclonal proteins in the blood, urine, and tissues with associated organ dysfunction [1]. The clinical onset of MM is often preceded by an asymptomatic premalignant condition called monoclonal gammopathy of undetermined significance (MGUS). MGUS, in turn, can evolve into smoldering MM (SMM), an intermediate phase in which PC expansion and gene mutations increase the risk of evolution to active MM [2,3,4]. MM exhibits broad heterogeneity in clinical presentation, molecular features, and treatment effectiveness [5]. Current MM therapies are based on a combination of conventional chemotherapy, corticosteroids, and one or more of the newer agents—such as proteasome inhibitors (i.e., bortezomib, carfilzomib, and ixazomib), checkpoint inhibitors, immunomodulating compounds (i.e., lenalidomide, thalidomide, and pomalidomide)—or biological therapies, including monoclonal antibodies (mAbs) (i.e., daratumumab and elotuzumab) and chimeric antigen receptor (CAR)-T cell therapy. The leading role of the BMME components in MM progression and heterogeneity suggests that further characterization of its specific activities may help identify key therapeutic targets and foster the development of new approaches that aim to reinforce the immune system.

The BMME includes a non-cellular compartment formed by extracellular matrix (ECM) proteins (laminin, fibronectin, and collagen) and soluble factors (cytokines, growth factors, and chemokines) and a rich cellular compartment constituting hematopoietic cells (myeloid cells, T lymphocytes, B lymphocytes, and natural killer (NK) cells) and non-hematopoietic cells (fibroblasts (FBs), osteoblasts, osteoclasts, endothelial cells (ECs), endothelial progenitor cells (EPCs), dendritic cells (DCs), pericytes, mesenchymal stem cells (MSCs), and mesenchymal stromal cells) [6]. In this specialized BM niche, all cells are protected from apoptotic stimuli and may, therefore, actively promote disease progression. Since the BM niche is the primary residence of long-lived PCs, the complex interaction among its cellular components, ECM proteins, and soluble factors may play a major role in the survival of malignant PCs [7].

The immune system acts as a critical rheostat that fine-tunes the balance between dormancy and disease progression in MM. Even if malignant PCs are not completely eliminated, the immune system is critical for maintaining functional dormancy at early stages; however, malignant PCs eventually evade immune control and foster progression toward active MM, in which dysfunctional effector lymphocytes, tumor-educated immunosuppressive cells, and soluble mediators act in coordination as a barrier against anti-MM immune response. An in-depth understanding of this dynamic process, known as “cancer immunoediting”, will provide important insights into the immunopathology of PC dyscrasias and, hopefully, help organize the most effective anti-MM immunotherapy [8]. 

Interestingly, a growing body of evidence suggests that a complex interaction between non-hematopoietic stromal cells and the BM immune system may display unique functions to support pro- and anti-tumor events in the BM niche, thus highlighting the relevant roles of immune components in the impaired anti-MM immune responses and disease progression [9].

Among BM immune cells, MSCs have long been recognized as key players in immune response, actively promoting the homing of PCs in the BM by secreting C-X-C Motif Chemokine Ligand 12 (CXCL12) (CXCR4 ligand), hence providing contact-dependent support for PCs by integrins and enhancing the secretion of pro-survival factors, such as interleukin-6 (IL-6) and vascular endothelial growth factor (VEGF). The reciprocal interactions between PCs and MSCs induce MM progression [10]. According to previous studies, MM-educated MSCs acquire the ability to produce high numbers of pro-inflammatory cytokines and growth factors that favor the accumulation and chemoresistance of malignant PCs [11,12]. These observations suggest that MM evolution might progressively define unique and complex immune phenotypes in the BM components, including bystander immune cells. 

Notably, immune changes represented by an increased number of terminal effector T cells and group 1 innate lymphoid cells can be observed from the MGUS stage to active MM [13]. Moreover, although stem-like tissue-resident T cells can still be detected in MGUS patients, subjects with advanced MM are characterized by the progressive loss of this T cell subset and the accumulation of senescent and exhausted T cells, suggesting that the T cell phenotype changes dynamically during disease progression [14]. Similarly, altered polarization of T cells, particularly T helper (Th) 17-skewed cells, has been reported in patients with active MM and associated with the increased release of IL-1, IL-6, and transforming growth factor-β (TGF-β) in the pro-inflammatory BM niche [15]. 

Despite some evidence pointing toward an accumulation of hypoproliferative, senescent CD571 and CD81 T cells in MM, the presence of exhausted T cells in newly diagnosed patients is controversial. Indeed, CD81 T cells from these patients rarely express high levels of multiple immune checkpoint receptors (i.e., programmed death-1 (PD-1), cytotoxic T-lymphocyte associated antigen-4 (CTLA-4), TIM-3, and LAG-3), which represents a cardinal feature of T cell exhaustion [16]. In addition, increased levels of PD-1 on T CD4^+^, which has been observed more in relapsed MM patients compared with MM and MGUS ones and is able to interact with PD-ligand 1 (PD-L1) on PCs and DCs, are correlated with MM progression [17].

Moreover, NK cells from MM patients display reduced expression of activating receptors and parallel upregulation of PD-1 receptors, the latter facilitating the inhibition of NK cytotoxicity by MM cells expressing higher levels of PD-L1 [18]. 

As professional antigen-presenting cells (APCs), DCs act as a link between innate and adaptive immunity. DCs from MM patients are also dysfunctional, are involved in PC survival, and may be included among the key determinants for the progression from MGUS to active MM [19]. 

The specific significance of different immune players in the BMME of MM patients is still an open field of investigation since the uncertain clinical responses to immune checkpoints inhibitors make it difficult to identify reliable predictive biomarkers. Next, we summarize the current knowledge on the importance of immunoediting in MM progression, focusing on the most common MM immune checkpoints identified so far and the relevant clinical/prognostic value of specific drug inhibitors.

## 2. Immunoediting in MM Progression

In MM, the “cancer immunoediting” is attributable to multiple factors, including the suppressive activity of tumor-associated macrophages (TAMs) and myeloid-derived suppressor cells (MDSCs), the immunotolerance toward malignant cell antigens, progressive T cell exhaustion, and the alteration in cytokine production [9,20,21]. The complex and dynamic rearrangement of all these elements occurs throughout three phases (elimination, equilibrium, and escape), whose combination fosters disease development [22].

The immunosuppression mechanism, reinforced by both tumor cells and the BMME components, involves lymphocyte effectors, immunosuppressive cells, and immunity-dampening molecules. With disease progression, somatic mutations in PCs can be immunogenic and induce neoantigen-specific NK and T cell activation. Concomitantly, the effector cells become silenced and gradually less effective, as confirmed by the increased numbers of neo-antigens found in relapsed MM patients compared with newly diagnosed subjects [23,24]. Compared with the MGUS stage, exhaustion and senescent signs of T cells appear more evidently in the advanced MM stage [13,14,25].

The elimination phase, which limits PC growth, matches with strong immune activity against PCs and subsists in a dynamic balance (the intermediate phase) with the escape stage. The equilibrium phase can last for a long time before switching to the escape phase; it is characterized by malignant PC dormancy, which may correspond to MGUS and/or SMM clinical disease appearance. Importantly, in patients undergoing medical therapies and autologous stem cell transplantation, the switch to the last stage is potentially a reversible process [9,26,27]. In this context, NK and CD8^+^ T cells are key actors in immunoediting evolution, and among the main mediators, perforin and interferon-γ (IFN-γ) as well as the adhesion receptor CD226 (DNAM-1) are responsible for the elimination process [28]. When triggered by its stress-induced ligands (Nectin2/CD112 and PVR/CD155), which are frequently over-expressed on malignant PC surfaces, the DNAM-1 receptor controls T and NK cell activation [29] and confers resistance to bortezomib and cyclophosphamide, contributing to slowed paraproteinemia and enhanced survival in mice models [28]. DNAM-1 also has a functional role in the NK-dependent killing of malignant PCs, strictly depending on the presence of Nectin-1 and PVR on their surfaces [30]. Compared with patients in remission and healthy controls, a reduced amount of DNAM-1 on CD56dim NK cells from patients with active disease was discovered, explaining the role of NK cells in MM pathogenesis [30]. Like CD226, the NK group 2D (NKG2D) receptor binds two stress-induced ligands (major histocompatibility complex class I-related chains A and B, MICA/B) and contributes to NK and T cell activity during the elimination phase, acting in combination with UL16-binding proteins (ULBPs) [31]. Overall, the data reveal that alterations in the NKG2D pathway are associated with the progression from MGUS to active MM [32], indicate that early changes in both innate and adaptive immunity are in place in MGUS-stage PC tumors (including an increasing number of T CD8^+^ and group 1 innate lymphoid cells), and indicate that a decrease in stem-like cells may underlie the loss of immune surveillance in MM progression [13]. 

The transition from MGUS to active MM also entails changes in inflammatory molecules. Indeed, the tumor secretion of phlogistic mediators, such as TGF-β, IL-10, IL-6, and prostaglandin E2 (PGE2), contributes to immunological imbalance and, combined with non-tumoral PC dysregulation, exposes MM patients to infections [33]. The numerical alteration and progressive impairment of the NK cell population toward the MM stage are related to TGF-β secretion, which leads to the defective release of INF-γ and suppresses antibody-dependent cellular cytotoxicity (ADCC) [34]. 

Further support for the role played by the BMME in MM progression comes from the contribution to the escape mechanism through the osteoclasts’ production of Gal-9 and proliferation-inducing ligand (APRIL), two signaling molecules promoting T cell apoptosis, and through PD-L1 expression in MM cells [35]. As previously mentioned, PD-L1 expression on malignant PCs has been associated with drug resistance, and serum levels of PD-L1 are predictive of worse progression-free survival (PFS) [36,37].

T and B regulatory cells (Tregs and Bregs), MDSCs, and TAMs all play relevant roles in MM immune escape by inhibiting the cytotoxic functions of T cells and NK cells and stimulating angiogenesis and proliferation, thereby promoting disease progression [38]. In the BMME, crosstalk between toll-like receptor (TLR)-2 and damage-associated molecular patterns (DAMPs) [39] is among the hypothesized mechanisms through which TAM precursors support MM progression [40]. In this context, S100A9, a fundamental DAMP that stimulates IL-18 and promotes MM progression by interacting with TLR-4 and RAGE [41], fosters a pro-inflammatory environment in the BM niche [41]. Moreover, TAM survival and differentiation in the BMME are favored by colony-stimulating factor-1 (CSF-1), whose increased levels are related to disease progression [42,43]. 

During the active phase of MM disease, the involvement of BMME in “cancer immunoediting” becomes more evident: specifically, MSCs upregulate IL-6 production and exhibit high levels of CD40/CD40L and adhesion molecules, such as VCAM-1, ICAM-1, LFA-3, junctional adhesion molecule-A (JAM-A), and human leukocyte antigen (HLA) system molecules (HLA-DR and HLA-ABC) [38]. Combined with MSC-derived microvesicles, all these elements contribute to immunoescape, drug resistance, and proliferation mechanisms. This concept is supported by recent findings showing that, by slowing down PCs’ uptake of microvesicles, integrin inhibitors (targeting α4β1 integrin and CD29 together) undermine malignant progression [44]. Similarly, a reduced inhibition of T cells proliferation with a shift in the Th17/Tregs balance when T cells were co-cultured with MM BM-MSCs [45] further supported the presence of immune dysfunction in both upstream mechanisms related to the dysregulated interactions of immune cells and the altered expression of downstream signaling by adhesion molecules and cytokines.

Lastly, the dormancy phase, starting from the early stage of disease, is embedded in MM progression and is a process involving genetic aberrations [46]. The dormancy status and its evolution in pre-cancerous lesions are tightly interconnected with the immune system and its balance, the BMME cells and their molecular components, and the genetic modifications of cancerous cells. Moreover, because of their role in homing PCs in BM niches, MSCs seem to be crucial to the survival of malignant PCs, drug resistance, and disease progression. MSCs and FBs ease the pathogenetic interaction between PCs and BMME via the excretion of CXCL12 (CXCR4 ligand, expressed on ECs and malignant PCs), which is relevant in the “cancer immunoediting” process [10,47]. The triggering of the CXCR4/CXCL12 signaling pathway enhances trans-endothelial migration and the homing and adhesion of cancer cells in the BM niche; it also increases the expression and excretion of other disease-progressing molecules (i.e., IL-6, integrins, and growth factors) [48,49]. Moreover, PCs expressing CD28 interact with plasmacytoid dendritic cells (pDCs) and the CD11c^+^ conventional dendritic cells (cDCs) CD80/CD86, triggering the former to excrete IL-6 and the latter to survive. CD28 pathway is regulated by a cDC, Treg, and PC cross-talking mechanism, and its signaling pathway is suppressed by CTLA-4 [50,51]. Furthermore, pDCs promote drug resistance, foster chemotaxis, and excrete high levels of IL-6 and IFN-γ [52]. They also contribute to the upregulation of kynurenine-3-mono-oxygenase. In the context of the BMME, this enzyme shifts the balance between tryptophan and kynurenine metabolites, abolishing the immune anti-tumoral response in both newly diagnosed MM patients and MM patients under treatment [53].

Interestingly, osteoblasts inside the BM niche allow malignant PCs to retain a quiescence state and stem-like characteristics. RANKL-mediated osteoclast activation (associated with disease progression and malignancy) importantly contributes to interrupting the cancer cells’ dormancy [54].

## 3. T Cell Dysfunction Occurs in MM Progression

Despite the role of other immune cells, CD4^+^ Treg and T cytotoxic CD8^+^ cells have emerged as the dominant effectors of host control for MM PCs (Figure 1). The progression from MGUS to active MM is associated with alterations in Tregs and terminal effector CD8^+^ T cells (TTEs) and is correlated with reduced survival in patients with recent MM diagnosis [1].

The clinical finding of an association between improved clinical outcome and reduced Tregs/Th17 cells ratios or Treg frequency and oligoclonal expansion of TTEs suggests that Tregs and the expansion of TTEs are key players in immune surveillance in MM [55]. This further entails that T cells may recognize MM PC antigens and differentiate into TTEs, which are able to exert cytotoxicity against PCs through perforin and granzyme expression. 

However, whether BM residency is necessary to confer tumor control is still unclear. Nevertheless, the same BM residency of immune cells can be considered the “Green Card” that allows permanent MM surveillance at the site of disease initiation and progression [56].

Tregs are a subset of CD4^+^ T lymphocytes characterized on the surface by the CD25^+^ CD127low phenotype and the expression of the transcription factor forkhead box P3 (FoxP3) [57]. If the homeostatic balance between Treg-mediated suppression and T effector cell activation is unbalanced in favor of effector activation, autoimmune disease emerges. In the case of malignancy, excessive Treg activity leads to the suppression or exhaustion of effector cells and a lack of tumor immune surveillance. Compared with MGUS, MM displays a skewing of the Treg and pro-inflammatory Th17 cell balance in favor of Tregs [58]. 

It has been postulated that Tregs are implicated in MM progression on the basis of their contribution to the complex immunosuppressive environment through the secretion of IL-10 and TGF-β by APRIL/TACI-dependent mechanisms and through the CD39/CD73 adenosine pathway and direct inhibition of effector T cell responses [59]. In particular, the secretion of IL-6, TGF-β, and IL-1β in the BM niche promotes Th17 production, inducing IL-17 release, which correlates with MM cell growth [60].

Treg variation between the elimination/equilibrium (MGUS) and the escape stage (MM) seems particularly intriguing, and, rather than a skewing of the balance between Tregs and pro-inflammatory Th17 in favor of Tregs, it might represent active Treg differentiation involving the regulation of ectonucleotidase CD39 expression and activation at BM residency [56]. Understanding changes in the Treg compartment holds the potential to improve our comprehension of the clinical stability in MGUS and MM progression, with relevant implications for the clinical diagnosis, prognosis, and successful therapeutic approaches.

## 4. Dendritic Cells as Important Players in MM Immune Response

Since DCs represent a bridge between the innate and adaptative immune responses, they are master APCs in all tissues, able to capture, process, and present tumor-(neo)antigens (Ags) to naïve T cells via major histocompatibility complex (MHC) molecules (Figure 2). Because of their capacity for cross-presenting Ags and inducing specific T cellular and B humoral immune responses, DCs are a promising tool for immunotherapy in MM [61].

As sentinel cells of the innate immune system, DCs are able to recognize endogenous danger molecules called DAMPs, which are released by damaged or dying cells via pattern recognition receptors (PRRs) on the cell surface. Subsequently, DCs secrete necessary cytokines allowing the activation of innate immune cells [62]. Simultaneously, DCs process and present these Ags on the cell surface, allowing immature DCs (imDCs) to switch to mature DCs (mDCs). The latter increase the expression of co-stimulatory molecules and immunostimulatory cytokines (CD80, CD86, and CD83, IL-12, IL-10, and tumor necrosis factor (TNF)) (Figure 2) [63]. In addition, imDCs conserve the MHC-II molecules in the late endosomal and lysosomal compartments, whereas in mDCs, the molecules are located on the cell surface. Upon Ag presentation via MHC molecules, mDCs migrate to draining lymph nodes in a chemokine-dependent manner. CCR7 and its cognate ligands (C-C motif ligand (CCL)-19 and CCL-21) allow the homing of DCs through the lymphatic vessels to the T lymphocyte-enriched zone in the secondary lymphoid organs. In the draining lymph nodes, mDCs trigger naïve T cells to differentiate into disparate T effector cells (i.e., Th1, Th2, Th17, T follicular helper (TFH) cells, Tregs, and CD8^+^ CTLs), resulting in specific T cell responses [64,65]. Depending on the Ag nature, by engaging the MHC-I presenting endogenous Ags and MHC-II presenting exogenous Ags, DCs elicit respectively adaptive CD8^+^ or CD4^+^ T cell immune responses. Here, a process called “cross-presentation” between exogenous Ags on MHC-I molecules results in CD8^+^ CTL activation. Interestingly, each DCs subset contributes differently to the immune response; for instance, cDC1s excel in cross-presenting exogenous Ags via MHC-I to CD8^+^ CTLs and secrete IL-12, thereby promoting Th1 responses [66].

Extensive literature data are available on the number, phenotypic profile, and functional status of DCs in MM progression. Specifically, the number of circulating DCs in healthy subjects includes 0.1–2.0% of the mononuclear cells [67], while a significant alteration is observed in MM patients, with approximately a 50% reduction in myeloid DCs (BDCA1^+^) and pDCs (BDCA2^+^) that is independent of the patient’s disease stage. Leone et al., found that during disease progression from MGUS to active/symptomatic MM, the myeloid DCs (CD11c^+^) and pDCs (CD11c^–^ CD123^+^) accumulate in the BM niche. This is paralleled by an increase in tumor burden, as both mDCs and pDCs exert immunosuppressive and tumor-promoting properties [68].

BMME immunological inhibitory cytokines induce phenotypic alterations and functional deficiencies, which include impaired DC differentiation, maturation, and activation. The most involved cytokines are TGF-β1, VEGF, IL-6, and IL-10. These factors can induce hyperactivation of STAT3 and extracellular signal-regulated kinase (ERK) pathways, which may be responsible for defective DC differentiation. TGF-β1 and IL-10 are both secreted by MM cells and play a significant role in deficient CD80/86 upregulation during DC maturation. In addition, the excessive production of TGF-β1 by MM cells suppressed allogeneic T cell responses and favored the differentiation and expansion of Tregs, resulting in tumor-associated immune tolerance [69]. While tumor-derived VEGF is engaged in the impaired DC function due to inhibitory effects on DC maturation and differentiation, it is also responsible for T cell exhaustion. Previous researchers confirmed the importance of MM cell adhesion to bone marrow stromal cells (BMSCs), which in turn secrete IL-6 in an NF-κB-dependent manner, supporting MM cell growth and survival. IL-6, which can also be secreted by MM cells, affects CD4^+^ T cell differentiation, inhibiting Th1 polarization and promoting Th2 differentiation. Furthermore, IL-6 stimulates CD34^+^ precursor cell differentiation into monocytes instead of DC progenitors through the upregulation of CD14 and the downregulation of CD1a, HLA-DR, CD40, and CD80 [70].

These large numbers of observations and experimental findings support the concept that MM cells are intelligent evaders of immunosurveillance and may employ a variety of mechanisms to disturb B cell immunity, promote Treg expansion, and suppress CTL activity while concomitantly directing their inhibitory activity on DC differentiation, maturation, and functions [70].

## 5. Immune Checkpoints and MM Progression

The high heterogeneity among MM patients has increased the need to identify immune checkpoints in the BMME that regulate MM physiopathology. These molecules represent the modulators of the signaling pathways responsible for immunological tolerance, a concept that prevents the immune system from destroying its own cells. In MM pathogenesis, recognition of biomarkers for the identification of patient populations that are likely to respond to therapy and/or have fewer side effects from therapy is needed. Accordingly, several factors that provide prognostic information and/or predict responses to checkpoint inhibitors have been identified. Immune tolerance is partly mediated by CTLA-4 and PD-1, two immunomodulatory receptors expressed on T cells that trigger inhibitory pathways dampening T cell activity. CTLA-4 and PD-1 immune checkpoints constitute the major immune escape mechanism in MM (Figure 2). CTLA-4 regulates T cell proliferation early in the immune response, primarily in the lymph nodes, and is more prominently expressed in patients with active MM compared with MGUS patients [71]. Instead, PD-1 suppresses T cells in the immune response, primarily in the peripheral tissues, and its expression in NK and T cells differs between relapsed/refractory MM patients and patients with MGUS or newly diagnosed MM [72]. The clinical profiles of immuno-oncology agents targeting these checkpoints may vary according to their mechanistic differences. 

### 5.1. CTLA-4

CTLA-4 is a member of the immunoglobulin superfamily and a negative regulator of T cell activation. The T cell receptor complex initially recognizes Ags; then, the binding of CD28 to CD80/CD86 on T cells and APCs, respectively, generates a primary positive co-stimulatory signal (Figure 2). After activation, CTLA-4 is expressed on T cells and exerts its negative regulatory effects by competing with CD28 for CD80/CD86 and blocking downstream pathway activation.

In MM T cells, CTLA-4 is upregulated and, via competitive bidding for the co-stimulatory molecules CD80/CD86, negatively modulates the activated T cells [73].

### 5.2. PD-1/PD-L1

PD-1 is a member of the B7/CD28 family of co-stimulatory receptors. It regulates T cell activation through the binding to PD-L1 and PD-L2 ligands (Figure 2). Similar to CTLA-4 signaling, the PD-1 binding inhibits T cell proliferation, production of IFN-γ, TNF-α, and IL-2, and reduces T cell survival. PD-1 expression is a hallmark of “exhausted” T cells that have experienced high levels of stimulation or reduced CD4^+^ T cell help [25]. This state of exhaustion, which occurs during chronic infections and cancer, is characterized by T cell dysfunction, resulting in suboptimal control of infections and tumors. 

In MM pathophysiology, PD-L1, which was first identified as B7 homolog-1 (B7-H1), is widely expressed on PCs and inhibits antitumor T cell responses associated with poor prognosis. PD-L1 expression levels are higher in MM PCs compared with those in MGUS patients and healthy PCs, and its expression is often upregulated upon relapse or in the refractory phase [74].

Interestingly, levels of soluble PD-L1 are elevated in the peripheral blood of newly diagnosed MM patients, and they are associated with a low response to treatment and shorter PFS [37]. PD-1 is overexpressed on T cells and NK cells in MM patients, and PD-1^+^ T cells are highly enriched in MM-specific effector cells [75]. Unfortunately, PD-1/PD-L1 interactions seem to undermine an effective anti-MM immune response and contribute to severe immune suppression and MM drug resistance. Accordingly, patients with an increased frequency of PD-1-expression on T cells after autologous stem cell transplant may present a higher risk of relapse [76]. Blockade of PD-1/PD-L1 enhances T cell and NK cell-mediated anti-MM responses in vitro and in vivo, and the administration of anti-PD-L1 or anti-PD-1 antibodies significantly decreases disease progression in MM mouse models [77].

## 6. Immunotherapy in MM

Conventional treatment of MM is age- and disease-stage-related. While SMM patients require only periodic observation, the active disease needs to be instantly treated. The conventional first-line therapy consists of the administration of thalidomide/lenalidomide, bortezomib, and dexamethasone. 

Autologous stem cell transplantation can be performed in addition to pharmacological treatment or singularly, depending on comorbidity and age-related factors [78].

A certain number of MM patients are treated with allogenic bone marrow transplantation (BMT) in association with maintenance drug administration. Because of the excretion of IL-17 by donor cells, IFN-γ is included in long-term therapy to avoid post-BMT relapse; lenalidomide is also part of the treatment because of its ability to ensure residual MM cell dormancy [79].

Results from immunotherapy studies suggest that, when administered separately, allogeneic BMT, immune checkpoint inhibitors, and DC-based vaccines have limited effects in a small number of patients (Figure 3) [38].

### 6.1. ImiDs and mAbs

Immunomodulatory drugs (ImiDs, including thalidomide and its analogs) promote cancer cells apoptosis, foster the proliferation and activity of NK and T cells (through cereblon-dependent degradation of the transcription factors IKZF1 and IKZF3) [80,81], improve the production of INF-γ and IL-2 by Th1 cells, enhance ADCC [38], and contain CD4^+^ and CD8^+^ IL-10 release, which enhances NK cell activation [82]. For treatment of refractory MM patient, these agents can be combined with mAb directed to specific targets: daratumumab, as well as isatuximab, by targeting CD38 on the MM cells surface, either in monotherapy or in combination with bortezomib and/or dexamethasone, is capable of promoting ADCC, apoptosis, complement-mediated cytotoxicity, antibody-dependent cellular phagocytosis (ADCP) and T cells response, targeting CD38 on the MM cells surface. In addition, this reduces MDSC, Treg, and Breg cell activity, leading to enhanced PFS [38]. It has also been observed that daratumumab depletes the CD38^+^ cell pool, resulting in an increased number of cytotoxic cells (Figure 3) [83]. The Dara-VTD regimen (daratumumab plus bortezomib, thalidomide, and dexamethasone) was approved in early 2020 by the Food and Drug Administration (FDA) and the European Medicines Agency (EMA) as a new induction therapy capable of improving the overall survival (OS) and response rate (RR), thus ensuring longer follow-ups [84]. Several other regimens with anti-CD38 Abs, in combination with carfilzomib and traditional chemotherapeutics, are currently being evaluated for first- and second-line and relapsed patients treatments [85,86]. Whether anti-CD38 mAb administration is compatible with CAR-T therapy is still an open question because of the presence of the antigen on activated T cells [87].

Elotuzumab targets SLAMF7 (CD319), improving NK cell function and ADCC, whether associated with lenalidomide and dexamethasone [88]. SLAMF7 is over-exhibited in MM patients with the chromosomal translocation t(4;14)(p16;q32) and is associated with very poor prognosis [89]. Several elotuzumab-enriched schemes of therapy have shown promising results in relapsed patients [88,90], and analogous outcomes are expected from an ongoing trial on a quadruple-drug induction and consolidation regime for newly diagnosed MM patients eligible for transplantation (HD6 trial, Elo-VRD; DSMMXVII trial, Elo plus carfilzomib, lenalidomide, and dexamethasone). An ongoing phase 3 trial on relapsed MM patients is investigating the combination of elotuzumab with anti-PD-1/anti-PD-L1 mAbs (NCT02726581).

Anti-IL17A mAb, in conjunction with PDR001 (an anti-PD-1 mAb), is currently being evaluated for the treatment of MM-relapsed patients. Preclinical studies have evaluated ulocuplumab, an anti-CXCR4 mAb, as well as olaptesed pegol (PEGylated mirror-image l-oligonucleotide capable of inhibiting CXCL12 signaling activity), as two possible strategies to limit the spread of PCs and MM progression [91,92].

Further studies have investigated the efficacy of natalizumab, an anti-VLA4 mAb used for the treatment of multiple sclerosis that binds α4 integrins, in order to prevent the interaction between ECM components, BM stromal cells, and malignant PCs; it has emerged that the drug slows tumor cell proliferation, VEGF secretion, and angiogenesis and strengthens the effects of bortezomib and dexamethasone [93].

### 6.2. Immune Checkpoints Inhibitors

PD-L1 plays a fundamental prognostic and progression role in MM pathogenesis [94]. It is over-exhibited on malignant PCs because of INF-γ and IL-6 activation of intracellular pathways (i.e., MEK/ERK) and induces both drug resistance and anti-apoptotic mechanisms (higher expression of Ki67 and BCL-2) [95]. Nivolumab and pembrolizumab (anti-PD-1 mAbs) show better performance in stable MM disease patients rather than in refractory ones when combined with pomalidomide and dexamethasone or radiotherapy regimes [96,97] (Figure 3). The association between ImiDs and anti-PD-1/PD-L1 mAbs has recently been discontinued by FDA since this combination could cause fatally excessive immune responses, such as autoimmune cardiomyopathy [38]. Conversely, several preclinical trials show promising results when anti-PD-1 mAbs are administered in monotherapy after transplantation and at an early disease stage [28,79,98,99]. A phase II study (NCT02681302, ClinicalTrials.gov) analyzing the combination of nivolumab and ipilimumab (noted for its Treg suppression activity in vivo) [100,101] (Figure 3) reported positive preliminary results in high-risk transplanted patients (both newly diagnosed and recurrent ones), although effectiveness is limited by concomitant severe immune-related adverse effects (65%). 

Recently, new potential target antigens have emerged. A preclinical trial investigated the effectiveness of elotuzumab plus an anti-CD137 agonist (4-1BB) mAb on the promotion of T cell proliferation and cytotoxicity in an early disease stage [102]. Unfortunately, a preclinical study on a group of recently transplanted patients observed that the anti-CD137 mAb treatment upregulated PD-1 and TIM-3 on CD8^+^ cells [103], thus suggesting that an anti-PD-1 mAb should be able to counterbalance the anti-CD137 upregulating effects [104].

T cell immunoreceptor with immunoglobulin and ITIM domains (TIGIT) works by competitively contrasting CD226 action [16] (Figure 3). It inhibits NK and CD8^+^ activation against cancer cells and interferes with TIGIT1 Treg activity and the T–DC interface [105]. Consequently, TIGIT blockade interrupts DC-derived IL-10 excretion, hindering the immune-escape process [98]. TIGIT represents an interesting target given its recurrent presence on BM CD8^+^ cells surface. Its blockade can reverse the T cell exhaustion process and improve disease control in MM patients and post-BMT patients [98]. However, given the unfavorable benefit–risk profile and higher toxicity revealed, immune checkpoint inhibitor trials, such as NCT02579863, have been put on hold by the FDA.

### 6.3. CAR-T Cells

CAR-T cell technology is used for the treatment of a few hematological neoplasias and has recently been approved in a certain clinical subset of refractory MM patients. Among all possible targets, the main target of the engineered T cells is B-cell maturation antigen (BCMA); other targets are CD19 [106], CD138 [107], isoform variant 6 of CD44 (CD44v6) [108], CD70, SLAMF7 [109], integrin β7, Igκ [110], and TGF-β [111], which are currently being investigated (Figure 3).

BCMA is frequently expressed in MM PCs and constitutes a promising target in refractory patients [112]. A phase I trial showed that, in heavily pre-treated patients with refractory and relapsed MM, the BCMA/CAR-T cell treatment idecabtagene vicleucel (ide-cel, also called bb2121) produced an OS rate of 85% (with a complete response in 45% of patients) [113,114]. A “real-life” study on belantamab mafodotin, a highly selective MM targeted therapy, enrolled a cohort of patients that received a median of eight prior lines of therapy and revealed an overall response rate (ORR) of 33%, very similar to the ORR reported in the DREAMM-2 trial [115]. Other phase I–II studies have confirmed the strong effectiveness of BCMA/CAR-T cell therapy, thus proposing it as the future first-line therapy in relapsed or refractory patients [114,116,117,118]. BCMA/CAR-T therapy has demonstrated a good safety profile, assuring a low incidence of neurotoxicity and cytokines release syndrome events compared with other CAR-T protocols used to treat B-cell lymphoma and leukemia [114,119].

Unfortunately, the majority of patients relapse in any case [117,118], suggesting the existence of tumor resistance and circumventing mechanisms. Among them, the possible BCMA downregulation or complete loss by PCs, the accelerated CAR-T cell life, and the limited strength conditions of T cells [114,120], especially in highly treated patients, are currently being investigated.

To date, countermeasures pursued to address the resistance mechanisms include the use of γ-secretase inhibitors, which increase BCMA cellular expression on PCs to the detriment of the soluble BCMA fragment capable of inhibiting CAR-T cell function [121]; the redefining of CAR-T cell manufacturing protocols to improve suitability [122]; and new CAR-T-cell composition and humanized targeting domains to lower the immune reaction against CAR-T cells and enhance engraftment and in vivo expansion [121,123,124].

### 6.4. Ab-Drug Conjugates (ADCs)

For MM patients who are refractory or suffering from high-impact adverse effects from CAR-T, ImiDs, and mAbs therapies, BCMA-specific ADCs could represent a suitable alternative treatment. Belantamab mafodotin is an ADC that binds specifically to BCMA, eliciting an antibody-dependent cytotoxic response and releasing the cytotoxic agent auristatin F. It has shown an acceptable safety profile, except for the occurrence of keratopathy (31%) and hematologic dyscrasias [125], and an OS rate of approximately 30% in phase II trial patients refractory to daratumumab, ImiDs, and proteasome inhibitors [126]. Considering that its toxicity profile appeared manageable in relapsed and refractory patients, belantamab mafodotin has been recently approved by EMA for adult MM patients who have received at least four prior therapies and were refractory to at least one proteasome inhibitor, one IMiD, and an anti-CD38 mAB, and for patients who have shown disease progression on the last therapy regimen [125].

### 6.5. Bispecific mAbs

Bispecific T cell engager mAbs (BiTEs) are able to target two different antigen-binding sites, CD3 (or other T cell receptors) and BCMA (or other tumor cell receptors) and have been proposed as new agents to promote immune response (Figure 3). They seem able to strongly enforce T cell engagement and activation independently of the T cell receptor recognition mechanism [127] and have demonstrated a proper response in heavily treated patients [128] in combination with ImiDs [129].

## 7. Final Remarks and Future Perspectives

Here, we reviewed the most important components involved in the tight interaction between MMPCs and the BMME during MM progression. Since malignant PCs depend on the BMME for their survival, an in-depth understanding of the BM niche structure might also provide clues for more effective immune-mediated control. 

Increasing knowledge has undoubtedly clarified that MM cells use several concomitant strategies to escape immune surveillance. While is undeniable that successful treatment approaches should prevent immune cells from becoming MM’s best friends, several gaps in the comprehension of the crosstalk between MM and BM immune cells are still present, and a number of key questions remain unanswered. 

In the search for the best combination treatment, one of the most compelling needs is to overcome drug resistance while allowing a sustainable therapy approach. The evolving process of cancer immunoediting shows that a treatment strategy that considers the BMME and clonal evolution is as important as treating the MM cells themselves. Unfortunately, because of the cumulative toxic and side effects of multi-drug treatments, the combined management with single molecule-driven drugs is still difficult to achieve, as highlighted by some recent clinical studies. In this respect, BCMA-based therapy may represent a promising strategy, supporting research on targets similar to BCMA in the future. 

At present, while several specific inhibitors for the BMME have been evaluated in preclinical studies, most of them are not available in clinical practice. Incorporating the molecular approaches related to diagnosis and risk stratification into the routine diagnostic workup of patients remains a necessary approach for the use of personalized, biologically based treatments for MM.

Overall, the therapeutic strategies described in this review underline the importance of a novel approach to fighting MM heterogeneity and further support the notion that the individual patient profile may contribute to developing specific immune microenvironment-based prognostic and predictive scores. From this point of view, the identification of deregulated mechanisms may also translate into immune biomarkers able to distinguish patients at higher risk of progression of aggressive disease and become the starting point for planning novel immunotherapeutic approaches to improve MM patients’ outcomes. 

## Figures and Tables

**Figure 1 jcm-11-02513-f001:**
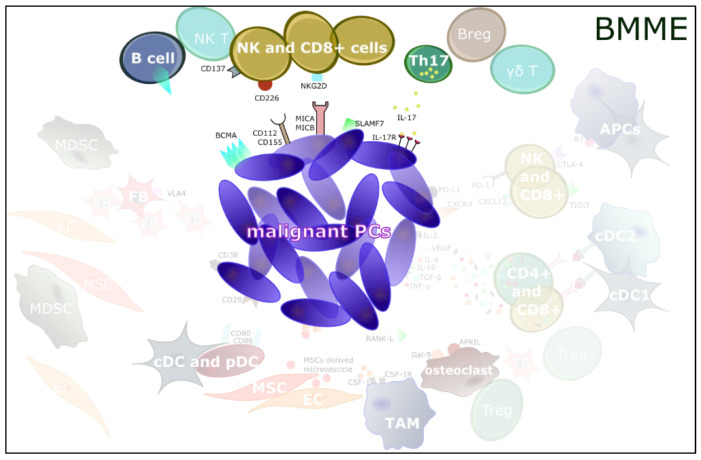
**The bone marrow microenvironment (BMME) in multiple myeloma (MM).** Complex interactions between non-hematopoietic stromal cells and BM immune system may support pro- and anti-tumor events in the BMME, highlighting the roles of immune components in the impaired anti-MM immune responses and disease progression. Innate and adaptive immune cells can recognize malignant plasma cells (PCs) and generate an anti-tumor immune response against tumors. A predominant role is attributed to immune cells, such as CD4^+^ Tregs and T cytotoxic CD8^+^ cells, which are considered effectors of host control for the MM PCs.

**Figure 2 jcm-11-02513-f002:**
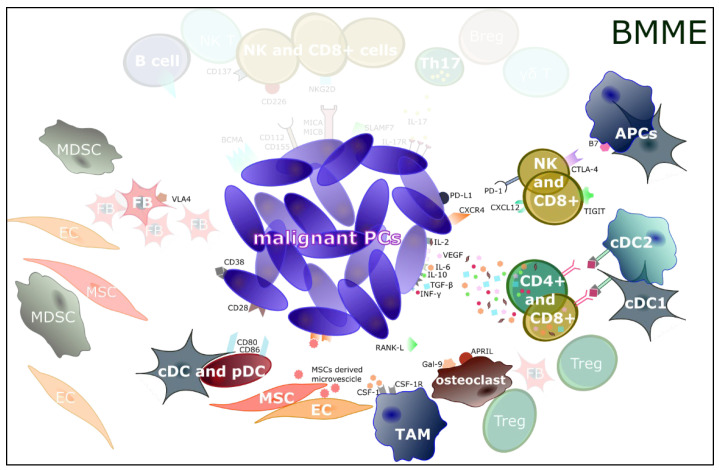
**The bone marrow microenvironment (BMME) in multiple myeloma (MM).** In BMME, plasmacytoid and conventional dendritic cells (pDCs and cDCs) play an important role in activating tumor-specific T cells with natural killer (NK), NKT phenotype, and gamma delta (γδ) T cells, inducing INF-γ production. Concomitantly, CD8^+^ and CD4^+^ cells realize an immunosuppressive milieu, producing transforming growth factor (TGF)-β, vascular-endothelial growth factor (VEGF), interleukin (IL)-10, IL-6, IL-17, and IL-2; interacting with antigen-presenting cells (APCs); and inducing T regulatory (Treg) cell differentiation and proliferation. The same interaction stimulates the CD28-CD80/CD86 contact and decreases the processing and presentation of tumor antigens, thus reducing malignant PC recognition by cytotoxic T CD8^+^ cells. Stromal cells, such as endothelial cells (ECs), fibroblasts (FBs), mesenchymal stem cells (MSCs), DCs, myeloid-derived suppressor cells (MDSCs), osteoclasts, and tumor-associated macrophages (TAMs), alongside immune cells, regulate different mechanisms (i.e., cell-to-cell adhesion; release of soluble factors, cytokines, chemokines, and growth factors) and activate several signaling pathways leading to MM progression. Among them, note that MSC-derived microvesicles are introjected by malignant PCs; B-cell maturation antigen (BCMA), CD38, and SLAMF7 are hyper-expressed on malignant PCs; osteoclasts highly activate and release RANK-L, Gal-9, and APRIL; VLA4 is expressed on FBs; CD137, CD226 (interacting with CD112 and CD155 on malignant PCs), and NKG2D (interacting with MICA\B on malignant PCs) are expressed on NK and CD8^+^ cells.

**Figure 3 jcm-11-02513-f003:**
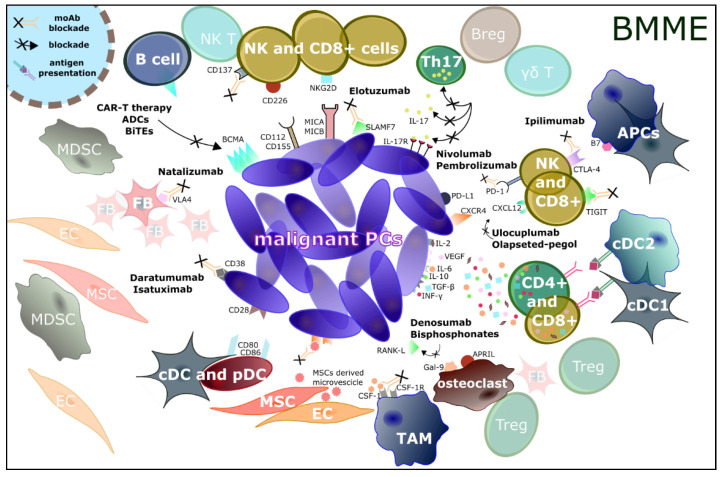
**Immunomodulatory drugs in MM.** Tumors have been shown to evade the immune system. This has led to the development of new agents to be used in combination with both well-established and innovative therapeutical schemes. Some of these immunological drugs include anti-CTLA-4 (ipilimumab), anti-CXCR4/CSCL12 system (ulocuplumab and olaptesed pegol), anti-PD-1/PDL-1 (nivolumab and pembrolizumab), and TIGIT, which counteract the blockade caused by immune checkpoints, enhance the immune response, and induce selective control on tumor growth, sometimes in the long term. As a result, the immune system is more active in recognizing the tumor as a foreign entity. CAR-T therapy is directed against the B-cell maturation antigen (BCMA) found on the surface of cancer cells; recognition and binding of BCMA lead to the proliferation of CAR-T cells, which can thus attack and kill the cancer cells expressing this antigen. Recent agents with the same target are Ab-drug conjugates (ADCs) and bispecific monoclonal Abs (such as BiTEs). A long list of mAbs capable of interfering with immunoediting pathways is currently approved, especially in combination regimes, and includes anti-CD38 (daratumumab and isatuximab), anti-VLA4 (natalizumab), and anti-SLAMF7 (elotuzumab) drugs; in addition, several new experimental compounds are currently under evaluation, such as CD137 blockers, MSC-derived microvescicle blockers, CSF-1/CSF-1R system blockers, and Th17/IL-17/IL-17R system blockers. Lastly, denosumab and bisphosphonates have been shown to be effective in slowing clinical disease progression and the immune escape process.

## Data Availability

All data generated or analyzed during this study are included in this published article.

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
