# Peer review of "The Leading Role of the Immune Microenvironment in Multiple Myeloma: A New Target with a Great Prognostic and Clinical Value"

_jcm, 2022, doi:10.3390/jcm11092513_

Round 1

Reviewer 1 Report

In this manuscript, authors Desantis et al have summarized the impact of immune microenvironment in multiple myeloma. A couple of points that the authors should work on to make the review easy to follow for the readers are:

  1. Since the aim of the authors is to provide a clinical perspective – the authors should mention which of the studies mentioned in the review were conducted with human patient samples or mouse models.
  2. The authors have Figure 1 to identify the different immune cells playing a role in MM. However, having a more interactive figure for each section (2-5), either individually or combined, indicating the mechanism of immune cells supporting MM would help readers grasp the concept and understand the topic better.
  3. The way the manuscript is currently written, it seems that the authors have stated one study after the other. It reads more like bullet points, especially sections 1 to 5, rather than a comprehensive writing. It is recommended that the authors connect thoughts for all the studies together for a more cohesive writing.
  4. To add value to the manuscript, the authors should consider including their own insight on the studies mentioned instead of just stating them as facts. For instance, how the study helped the field, how did it fill the gap in knowledge, what else is unknown and needs to be studied with regards to clinical relevance.
  5. Figures need to be cited in-text.

Author Response

In this manuscript, authors Desantis et al have summarized the impact of immune microenvironment in multiple myeloma. A couple of points that the authors should work on to make the review easy to follow for the readers are:

Q1. Since the aim of the authors is to provide a clinical perspective, the authors should mention which of the studies mentioned in the review were conducted with human patient samples or mouse models.

A1. To satisfy the Reviewer's request, authors have specified, at various points in the text, the nature of the studies, whether they are conducted in human patient samples or mouse models (e.i., line 155, line 158).

Q2. The authors have Figure 1 to identify the different immune cells playing a role in MM. However, having a more interactive figure for each section (2-5), either individually or combined, indicating the mechanism of immune cells supporting MM would help readers grasp the concept and understand the topic better.

A2. According to Reviewer’s suggestion, authors have added new Figures in the text, splitting the Figure 1. The new Figure 1 is associated to the sections 2 and 3 and the new Figure 2 is associated to the sections 4 and 5.

Figure legends have been modified accordingly.

Q3. The way the manuscript is currently written, it seems that the authors have stated one study after the other. It reads more like bullet points, especially sections 1 to 5, rather than a comprehensive writing. It is recommended that the authors connect thoughts for all the studies together for a more cohesive writing.

A3. We thank the Reviewer for having pointed this out. We apologize for the unclear description of these paragraphs. Based on his/her suggestions, the text from line 148 to line 456 has been revised.

Q4. To add value to the manuscript, the authors should consider including their own insight on the studies mentioned instead of just stating them as facts. For instance, how the study helped the field, how did it fill the gap in knowledge, what else is unknown and needs to be studied with regards to clinical relevance.

A4. To comply with this Reviewer’ s suggestion, we have expanded the conclusions by adding our considerations and insights, as well as by pointing out still unclear and critical aspects on this topic.

page 15, lines 645-663

[…] Increasing knowledge has undoubtedly clarified that MM cells use several concomitant strategies to escape immune surveillance. While is undeniable that successful treatment approaches should avoid immune cells to become MM’s best friends, several gaps in the comprehension of the crosstalk between MM and BM immune cells are still present, and a number of key questions remain unanswered.

In the search for the best combination treatment, one of the most compelling needs is to overcome drug resistance while allowing a sustainable therapy approach. The evolving process of cancer immunoediting teaches that a treatment strategy that considers the BMME and clonal evolution is as much important as to treat the MM cells themselves. Unfortunately, due to cumulative toxic and side effects of multi-drug treatments, the combined management with single molecule-driven drugs is still difficult to achieve, as highlighted by some recent clinical studies. On this respect, BCMA-based therapy may represent a promising strategy, supporting the research toward targets similar to BCMA in the future.

At present, while several specific inhibitors for the BMME have been evaluated in preclinical studies, most of them are not available in clinical practice. Incorporating the molecular approaches related to diagnosis and risk stratification into the routine diagnostic workup of patients remains a necessary approach for the use of personalized, biologically based treatments in MM.

Q5. Figures need to be cited in-text.

A5. We thank the Reviewer for having pointed this out. We apologize for this dearth. Based on the Reviewer’s comment, the Figures have been cited in the text.

Reviewer 2 Report

The authors have done a commendable job in reviewing the immune microenvironment in multiple myeloma. Following are my suggestions to improve the article.

1- Please consider adding immune exhaustion, immune escape, and immunotherapy in the keywords.

2- In line 41, use monoclonal proteins instead of monoclonal components.

3- Line 46, gene mutations increase the risk of developing active MM from smoldering myeloma is not very well established. Can we add more references to this fact?

4- As per ESMO, or NCCN guidelines, conventional chemotherapy is rarely used in upfront treatment of MM, and immune checkpoint inhibitors have not demonstrated efficacy in MM space. (https://pubmed.ncbi.nlm.nih.gov/27269947/) and actually had higher toxicity (https://ascopubs.org/doi/abs/10.1200/JCO.2018.36.15_suppl.8010) and these trials were put on hold by FDA.

5- Line 93, can you provide a reference for MSC and MM interaction resulting in immunosuppression?

6- Line 153,  "When triggered by its stress-induced ligands (CD112 and CD155), that are frequently over-expressed on malignant PCs surface, the DNAM-1 receptor controls T and NK cells activation and contributes to slow down paraproteinemia and enhanced survival", is there any mechanism to explain this?

7- Reference 28 doesn't seem to support the statement of mechanism of infections.

8- Line 366, CTLA4 and PD1 provide immune tolerance, will not call it immunosuppression. 

9- The mechanism of PD-1 and CTLA4 are described in too much detail and could be shortened for making paper concise. 

10-Line 422, please remove the word "heavily"

11- In the immunomodulatory drugs section, will recommend starting with daratumumab, as elotuzumab is one of the options but is inferior to daratumumab in utility. 

12- Line 504 needs restructuring. The trial of combination of IMiD and PD-1 inhibitors has been suspended by FDA due to excessive immune activation. 

13- Line 537- Please describe the time point for OS and CR for Idecel. At median follow up of 13.8 months, the CR rate was 33% and OS was 78%. Also please describe ORR, similarly describe these for Blanatamab. 

Author Response

The authors have done a commendable job in reviewing the immune microenvironment in multiple myeloma. Following are my suggestions to improve the article.

Q1. Please consider adding immune exhaustion, immune escape, and immunotherapy in the keywords.

A1. We thank the Reviewer for making us notice this. Keywords have been modified consequently:

page 1, lines 34-35

[…] Multiple myeloma; bone marrow niche; immune escape; immune exhaustion; immune checkpoint inhibitors; immune microenvironment; immunotherapy.

Q2. In line 41, use monoclonal proteins instead of monoclonal components.

A2. We thank the Reviewer for having pointed this out. The text has been modified accordingly:

page 1, line 41

[…] The abnormal and uncontrolled proliferation of PCs translates into accumulation of monoclonal components proteins (CM) in blood, urines and tissues with associated organ dysfunctions [1].

page 14, line 616

[…] CM            Monoclonal components

Q3. Line 46, gene mutations increase the risk of developing active MM from smoldering myeloma is not very well established. Can we add more references to this fact?

A.3 We thank the Reviewer for his/her valuable comment. In this regard, recent investigations have shown that the technique of WGS (Whole-genome sequencing) associated with multi-parameter flow-cytometry sorting represent the most comprehensive approach to characterize MM and myeloma precursor conditions (MGUS and SMM), due to their ability to describe the full genomic landscape of normal tissue from a few thousand cells, by defining genomic alterations such as single nucleotide variants, mutational signatures, copy number variants and structural variants. Thanks to this methodology it has been demonstrated that clinically stable cases of MGUS and SMM are characterized by a different genomic landscape and by differences in the temporal acquisition of MM defining genomic events associated with different conditions of disease progression.

According to the Reviewer’s comment, authors have added in the text the references concerning this topic (page 2, line 46):

  • Oben, B.; Froyen, G.; Maclachlan, K.H.; Leongamornlert, D.; Abascal, F.; Zheng-Lin, B.; Yellapantula, V.; Derkach, A.; Geerdens, E.; Diamond, B.T.; et al. Whole-Genome Sequencing Reveals Progressive versus Stable Myeloma Precursor Conditions as Two Distinct Entities. Nat Commun 2021, 12, 1861, doi:10.1038/s41467-021-22140-0.
  • Boyle, E.M.; Deshpande, S.; Tytarenko, R.; Ashby, C.; Wang, Y.; Bauer, M.A.; Johnson, S.K.; Wardell, C.P.; Thanendrarajan, S.; Zangari, M.; et al. The Molecular Make up of Smoldering Myeloma Highlights the Evolutionary Pathways Leading to Multiple Myeloma. Nat Commun 2021, 12, 293, doi:10.1038/s41467-020-20524-2.

All references have been updated accordingly.

Q4. As per ESMO, or NCCN guidelines, conventional chemotherapy is rarely used in upfront treatment of MM, and immune checkpoint inhibitors have not demonstrated efficacy in MM space. (https://pubmed.ncbi.nlm.nih.gov/27269947/) and actually had higher toxicity (https://ascopubs.org/doi/abs/10.1200/JCO.2018.36.15_suppl.80 10) and these trials were put on hold by FDA.

A4. Thank you for making us notice this. The text has been enriched with:

page 14, lines 585-587

[…] post-BMT patients [93]. However, given the unfavorable benefit-risk profile and higher toxicity revealed, immune checkpoint inhibitors trials such as NCT02579863 have been put on hold by the FDA.

Q5. Line 93, can you provide a reference for MSC and MM interaction resulting in immunosuppression?

A5. We thank the Reviewer for this comment. We have included references (page 2, line 91) about MSC and MM crosstalk resulting in immunosuppression, as suggested:

  • Botta C, Mendicino F, Martino EA, Vigna E, Ronchetti D, Correale P, Morabito F, Neri A, Gentile M. Mechanisms of Immune Evasion in Multiple Myeloma: Open Questions and Therapeutic Opportunities. Cancers (Basel). 2021 Jun 28;13(13):3213. doi: 10.3390/cancers13133213. PMID: 34203150.
  • Nakamura K, Smyth MJ, Martinet L. Cancer immunoediting and immune dysregulation in multiple myeloma. Blood. 2020 Dec 10;136(24):2731-2740. doi: 10.1182/blood.2020006540. PMID: 32645135.

All references have been updated accordingly.

Q6. Line 153, "When triggered by its stress-induced ligands (CD112 and CD155), that are frequently over-expressed on malignant PCs surface, the DNAM-1 receptor controls T and NK cells activation and contributes to slow down paraproteinemia and enhanced survival", is there any mechanism to explain this?

A6. We thank the Reviewer for his/her valuable comment. Sentences have been now rephrased to better clarify the described mechanism. Changes are reported below:

page 4, lines 151-159

[…] When triggered by its stress-induced ligands (Nectin2/CD112 and PVR/CD155), that are frequently over-expressed on malignant PCs surface, the DNAM-1 receptor controls T and NK cells activation [29], and contributes confers resistance to bortezomib and cyclophosphamide contributing to slow down paraproteinemia and enhanced survival in mice models [28]. DNAM-1 has a functional role also in the NK-dependent killing of malignant PCs, strictly depending on the presence of Nectin-1 and PVR on their surfaces [30]. Compared to patients in remission and healthy controls, a reduced amount of DNAM-1 on CD56dim NK cells from patients with active disease was discovered, explaining the role of NK cells in MM pathogenesis [30].

References have been updated accordingly.

Q7. Reference 28 doesn't seem to support the statement of mechanism of infections.

A7. We thank the Reviewer for his/her comment. All text has been modified accordingly:

page 4, lines 173

[…] Besides cell membrane antigens, changes in inflammatory molecules are also reported. The tumor secretion of phlogistic mediators such as TGF-β, IL-10, IL-6 and prostaglandin E2 (PGE2) contributes to immunological imbalance and, combined with non-tumoral PCs dysregulation, exposes MM patients to urinary tract infections and lung infections [28].

Q8. Line 366, CTLA4 and PD1 provide immune tolerance, will not call it immunosuppression.

A8. Text has been revised accordingly (page 10, line 406-407).   

Q9. The mechanism of PD-1 and CTLA4 are described in too much detail and could be shortened for making paper concise.

A9. We agree with Reviewer’s comment. The paragraph has been now modified (pages 10-11, from line 398 to line 456).

Q10. Line 422, please remove the word "heavily".

A10. According to the Reviewer suggestion, the text has been revised as indicated below:

page 11, lines 465

[…] A certain amount of MM patients is heavily treated with allogenic bone marrow transplantation (BMT) associated to maintenance drugs administration.

Q11. In the immunomodulatory drugs section, will recommend starting with daratumumab, as elotuzumab is one of the options but is inferior to daratumumab in utility.

A11. Based on the Reviewer’s suggestions, the text has been revised as indicated below:

pages 12-13, from line 499 to line 553

[…]: daratumumab, as well as isatuximab, by targeting CD38 on the MM cells surface, either in monotherapy or in combination with bortezomib and/or dexamethasone, is capable of promoting ADCC, apoptosis, complement-mediated cytotoxicity, antibody-dependent cellular phagocytosis (ADCP) and T cells response, targeting CD38 on the MM cells surface. In addition, it reduces MDSCs, Tregs and Bregs cells activity, leading to an enhanced PFS [38]. It has also been observed that daratumumab depletes CD38+ cells pool, resulting in an increased number of cytotoxic cells (Figure 3) [83]. The Dara-VTD regimen (daratumumab plus bortezomib, thalidomide and dexamethasone) was approved in early 2020 by the Food and Drug Administration (FDA) and European Medicines Agency (EMA) as a new induction therapy, capable of improving the overall survival (OS) and response rate (RR), thus ensuring longer follow-ups [84]. Several other regimens with anti-CD38 Abs, in combination with carfilzomib and traditional chemotherapeutics, are currently being evaluated for first- second-line and relapsed patients treatments [85,86]. Whether the anti-CD38 mAbs administration is compatible with CAR-T therapy is still an open question, due to the presence of the antigen on activated T cells [87].

 eElotuzumab that targets SLAMF7 (CD319) thus improving NK cells function and their ADCC, whether associated with lenalidomide and dexamethasone [88]. SLAMF7 is over exhibited in MM patients with chromosomal translocations t(4;14) (p16;q32), and is associated with very poor prognosis [89]. Several elotuzumab enriched schemes of therapy have shown promising results in relapsed patients [88,90], and analogous outcomes are expected from an ongoing trial on a quadruple-drug induction and consolidation regime, related to newly diagnosed MM patients eligible for transplantation (HD6 trial, Elo-VRD; DSMMXVII trial, Elo plus carfilzomib, lenalidomide, and dexamethasone). An ongoing phase 3 trial on relapsed MM patients investigates the combination of elotuzumab with anti-PD-1/anti-PD-L1 mAbs (NCT02726581).

Daratumumab, as well as isatuximab, by targeting CD38 on the MM cells surface, either in monotherapy or in combination with bortezomib and/or dexamethasone, is capable of promoting ADCC, apoptosis, complement-mediated cytotoxicity, antibody-dependent cellular phagocytosis (ADCP) and T cells response, targeting CD38 on the MM cells surface. In addition, it reduces MDSCs, Tregs and Bregs cells activity, leading to an enhanced PFS [38,85]. It has also been observed that daratumumab depletes CD38+ cells pool, resulting in an increased number of cytotoxic cells [86].

The Dara-VTD regimen (daratumumab plus bortezomib, thalidomide and dexamethasone) was approved in early 2020 by the Food and Drug Administration (FDA) and European Medicines Agency (EMA) as a new induction therapy, capable of improving the overall survival (OS) and response rate (RR), thus ensuring longer follow-ups [87]. The activity of an anti-PD-1/anti-PD-L1 mAb may have a synergic effect with daratumumab in relapsed patients (NCT02431208, NCT01592370, NCT03357952). Several other regimens with anti-CD38 Abs, in combination with carfilzomib and traditional chemotherapeutics, are currently being evaluated for first- second-line and relapsed patients treatments [88,89]. Whether the anti-CD38 mAbs administration is compatible with CAR-T therapy is still an open question, due to the presence of the antigen on activated T cells [90].

Anti-IL17A mAb, in conjunction with PDR001 (an anti-PD-1 mAb), is currently being evaluated for the treatment of MM relapsed patients. Preclinical studies have evaluated ulocuplumab, an anti-CXCR4 mAb, as well as olapseted-pegol (PEGylated mirror-image l-oligonucleotide capable of inhibiting CXCL12 signalling activity), as two possible strategies to limit the spread of PCs and MM progression [91,92].

Further studies have investigated the efficacy of natalizumab, an anti-VLA4 mAb used for the treatment of multiple sclerosis, which binds α4 integrins in order to prevent the interaction between ECM components, BM stromal cells and malignant PCs; it has emerged that the drug slows down tumor cells proliferation, VEGF secretion, angiogenesis, and strengthens bortezomib and dexamethasone effect. The selective adhesion molecule inhibitor Natalizumab decreases MM cell growth in the BMME: therapeutic implications [93].

References have been updated accordingly.

Q12. Line 504 needs restructuring. The trial of combination of IMiD and PD-1 inhibitors has been suspended by FDA due to excessive immune activation.

A12. We thank the Reviewer for his/her valuable comments. We agree on the need for revisiting the text:

page 13, lines 535-536

[…] The activity of an anti-PD-1/anti-PD-L1 mAb may have a synergic effect with daratumumab in relapsed patients (NCT02431208, NCT01592370, NCT03357952).

Q13. Line 537- Please describe the time point for OS and CR for Idecel. At median follow up of 13.8 months, the CR rate was 33% and OS was 78%. Also please describe ORR, similarly describe these for Belantamab.

A13. We provided to add the requested parameters as suggested:

page 14, lines 599-602

[…] A “real-life” study on Belantamab mafodotin, a highly selective MM targeted therapy, enrolling a cohort of patients that received a median of eight prior lines of therapy, revealed an overall response rate (ORR) of 33%, behaving in much the same way of the ORR reported in the DREAMM-2 trial [115].

A new reference has been added and all references have been updated accordingly.

Round 2

Reviewer 1 Report

The authors have addressed all the comments and the manuscript reads significantly better.

Reviewer 2 Report

Thank you for making the changes. Will recommend accepting the article with changes.